# How ready is the health care system in Northeast India for surgical delivery? A mixed-methods study on surgical capacity and need

Amrit Virk[1]*, Rebecca King[2], Michael Heneise[3], Lanuakum Aier[4], Catriona Child[4], Julia Brown[5], David Jayne[5], Tim Ensor[2]

1 Global Health Policy Unit, School of Social and Political Science, University of Edinburgh, Edinburgh, United Kingdom, 2 Nuffield Centre for International Health and Development, University of Leeds, Leeds, West Yorkshire, United Kingdom, 3 Faculty of Humanities, Social Sciences and Teacher Education, Department of Archaeology, History and Religious Studies, UiT The Arctic University of Norway, Tromsø, Norway, 4 The Highland Institute, Kohima, Nagaland, India, 5 School of Medicine, University of Leeds, Leeds, West Yorkshire, United Kingdom

☯ These authors contributed equally to this work.
* avirk@ed.ac.uk

**Data Availability Statement:** All relevant data are within the manuscript and its Supporting Information files.We can upload the data to the

## Abstract

### Background

Surgical services are scarce with persisting inequalities in access across populations and regions globally. As the world's most populous county, India's surgical need is high and delivery rates estimated to be sub-par to meet need. There is a dearth of evidence, particularly sub-regional data, on surgical provisioning which is needed to aid planning.

### Aim and method

This mixed-methods study examines the state of surgical care in Northeast India, specifically health care system capacity and barriers to surgical delivery. It involved a facility-based census and semi-structured interviews with surgeons and patients across four states in the region.

### Results

Abdominal conditions constituted a large portion of the overall surgeries across public and private facilities in the region. Workloads varied among surgical providers across facilities. Task-shifting occurred, involving non-specialist nursing staff assisting doctors with surgical procedures or surgeons taking on anaesthetic tasks. Structural factors dis-incentivised facility-level investment in suitable infrastructure. Facility functionality was on average higher in private providers compared to public providers and private facilities offer a wider range of surgical procedures. Facilities in general had adequate laboratory testing capability, infrastructure and equipment. Public facilities often do not have surgeon available around the clock while both public and private facilities frequently lack adequate blood banking. Patients' care pathways were shaped by facility-level shortages as well as personal preferences influenced by cost and distance to facilities.

Research Data Leeds Repository and make a DOI or URL available.

**Funding:** DJ, JB This research was funded by the National Institute for Health Research (NIHR) (16/137/44) using UK aid from the UK Government to support global health research. The funders had no role in study design, data collection and analysis, decision to publish, or preparation of the manuscript.

**Competing interests:** The authors have declared that no competing interests exist.

## Discussion and conclusion

Skewed workloads across facilities and regions indicate uneven surgical delivery, with potentially variable care quality and provider efficiency. The need for a more system-wide and inter-linked approach to referral coordination and human resource management is evident in the results. Existing task-shifting practices, along with incapacities induced by structural factors, signal the directions for possible policy action.

## Introduction

Conditions treatable by surgery account for nearly a third of the global disease burden, making it a critical area of public health and policy attention [1]. The investment case for surgery is evident—essential surgical procedures and anaesthesia, most of which could be provided at first-level hospitals, are a cost-effective undertaking with substantial returns on investment, preventing as many as 6–7% of all avoidable deaths in low- and middle-income countries (LMICs) [2]. Yet, a staggering 5 billion people—approximately 5 out of 7 people globally—lack access to safe, timely, and affordable surgical care [3–5]. Moreover, fewer than 6% of all surgical operations occur in LMICs, where over a third of the world's population lives [6]. The figures reveal vast geographical disparities, 295 in LMICs against 23,000 surgeries per 100,000 population in high-income countries, driven by shortages of medical staff, poor access to health care services, and weak record-keeping [6]. Hence, surgical resources need to be urgently scaled up in pursuit of achieving universal health coverage (UHC) to ensure everyone receives the good-quality, affordable health care services that they need [7]. Three procedures, caesarean section, fracture repair, and laparotomy, together known as the Bellwether Procedures, can treat the vast majority of surgical problems; ensuring they are available 24/7 at all first-level hospitals is a key strategy to expand surgical access [8]. A system's capacity to safely conduct these procedures also indicates its capacity to take care of the majority of first-level procedures (for example, if a laparotomy can be performed, then abscess drainage, circumcision, and chest tube placement can also be conducted safely). The Bellwethers, thus, act as "proxy" or indicator surgical procedures.

Estimates of the level of surgical care in India indicate substantial gaps in meeting required needs. Notably, a recent study estimated a requirement for 3646 surgeries annually per 100,000 population [9]. Astonishingly, as many as 90% of people in rural India lack access to safe surgical care [10], and current rates of surgical procedures are far below global averages (50–499 surgeries per 100,000) [11]. Within India, the health system in the North-Eastern states is one of the most underdeveloped, with poor access and health indicators [12]. This region, consisting of eight states, comprises a substantive 3.4% of India's population and 8% of the country's area according to the 2011 census. It is also distinct in Asia, with its characteristic ethnolinguistic diversity indicating the interaction of many vernacular health systems [13]. Located in the lower Himalayas region, the region's shared borders with other countries (in red on Map 1) as well as protracted conflict make it politically sensitive and limit research resources. Moreover, seasonal factors affect access to surgical services as road travel is very difficult or not possible during the period of heavy rains (Monsoon, May- July) in several locations across Northeast India. Opportunities for evacuating patients at reasonable cost by helicopter may not be possible during bad weather or in case of an unscheduled emergency. This factor affects readiness to access surgical services and should be described.Moreover, at both the health system and community levels, a lot remains unknown about the structures,

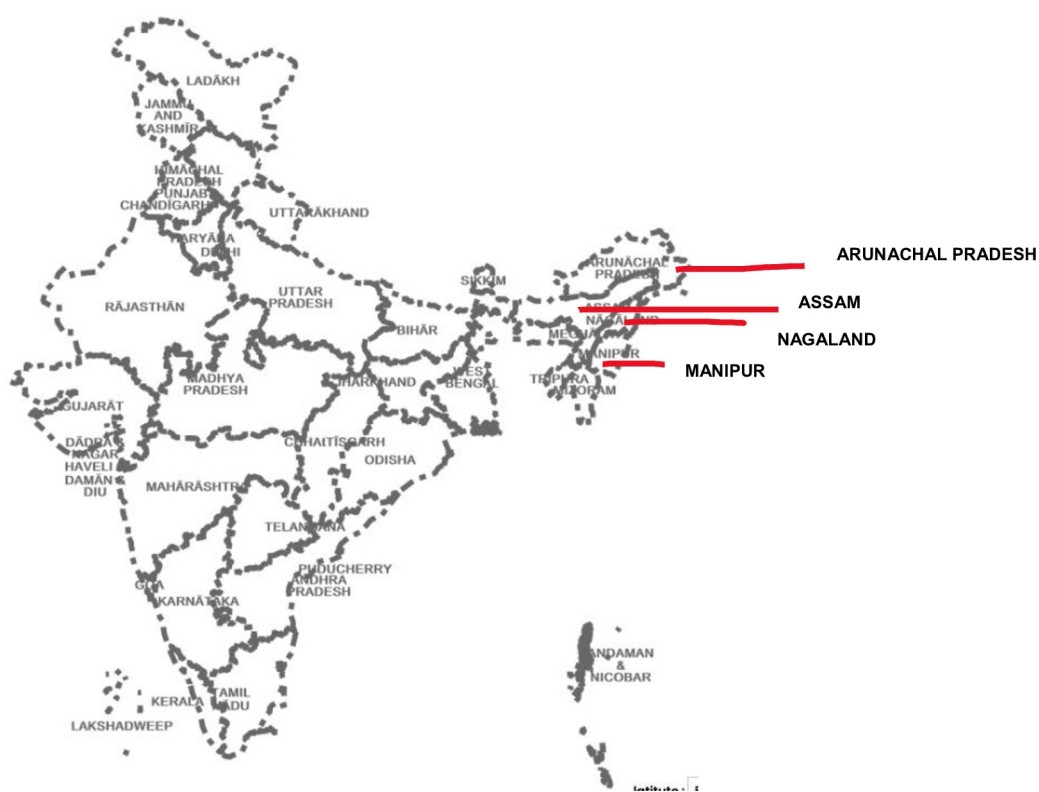

**Map 1. Study location (in red).** Source: Image:India-locator-map-blank.svg.

processes, and practices for surgical delivery and care-seeking in rural and NE India. An understanding of system readiness and the potential to provide basic surgical care is vital to inform technological and policy interventions to improve surgical care in rural India.

Panel 1: NIHR Global Health Research Group- Surgical Technologies

> This study is embedded within a large, multi-disciplinary project involving social science, engineering, statistical, and clinical expertise to develop low-cost surgical technology to address unmet surgical needs in under-resourced settings in Northeast India. The project focused on abdominal surgery as a significant component of overall surgical demand in the country. A mixed-methods study was designed to generate contextual information on health system capacity and care-seeking behaviours to inform the work of the other components to develop relevant clinical and engineering interventions. The UK-based team worked in close collaboration with regional institutions, including a clinician-led, humanitarian organisation providing medical care and surgical training in the region and a team of social scientists from the region.

## Aim

This mixed-methods study aimed to examine the state of surgical care in North Eastern India, namely health care system capacity and barriers to surgical care. It thereby developed a baseline for understanding the readiness and potential of facilities to deliver surgical care. This

study addresses a vital gap in current literature [14], by providing sub-national evidence on the current state of surgical care in India.

## Materials and methods

### Study design

This mixed-methods study involved qualitative interviews and health facility surveys conducted across four (of eight) states in NE India: Arunachal Pradesh, Assam, Nagaland, and Manipur (Map 1) between April 2019 and May 2020, together describing a substantive geographic area. The facility survey was designed to assess the health care system's readiness to provide basic surgical care and abdominal surgery. Semi-structured interviews focused further on surgical provisioning and patient care-seeking behaviours to uncover barriers to surgical delivery and patterns of care-seeking. A preliminary review of relevant secondary literature, national plans, and country reports accompanied this assessment of the existing state of surgical care in NE India.

The survey tool (Panel 2) and interview guides were drafted following a series of meetings among a multi-disciplinary team based in the UK and local researchers from India (surgeons, economists, anthropologists, statisticians, and social scientists). The study premise was to ensure that local stakeholders led the work on the ground and to build research capacity to undertake similar global health work independently. Hence, members of the UK team (AV, TE) worked closely with the local team (including MH, LA, and CC) to refine the topic guides and carry out the pilot interviews. Interview tools, including topic guides, were prepared in both English and Hindi and back-translated to check for accuracy. The local team conducted a few interviews in Nagamese (Nagaland), Assamese (Assam), and Meitei (Manipur), consulting a translator for Meitei. The survey tool and topic guides for each respondent category were further refined after initial pilot testing in the field. Ethical approval was obtained in-country from Sigma Research and Consulting (India) and the University of Leeds School of Medicine Research Ethics Committee (UK).

### Site selection

Convenience and purposive site selection led to a focus on four study districts (Table 1) where local doctors trained in a novel surgical technique were located (Panel 1). Local informants with long-standing clinical and research knowledge of the region assisted with site selection to ensure a comparison between larger urban districts (Dimapur) and rural locales. The facility survey was a census of all surgical facilities in these four districts, and interviewees were also located here.

Supported by UK- and India-based senior researchers, a 7-member team of local research assistants carried out the facility surveys and interviews between May 2019 and January 2020. The qualitative study methods are reported based on the standards for reporting qualitative research (SRQR) guidelines [15].

### Semi-structured interviews

This paper reports on interviews with 8 clinicians and 11 patients across 4 states (Table 1). Participants at surveyed facilities were recruited between May and November 2019 through purposive and snowballing methods to include those with first-hand knowledge and experience of surgical delivery and use. (Table 1). Additionally, provider amenability/use of less invasive techniques mainly laparoscopy (given concomitant cost and clinical benefits compared to open surgery) were also specifically sought to consider options for improving surgical delivery.

**Table 1. Sample of respondents.**

| State | District | No. of doctors interviewed | No. of patients interviewed |
|---|---|---|---|
| Arunachal Pradesh | Bomdila/Bhaloukpong | 1 | 0 |
| Assam | Chirang/Bongaigaon | 2 | 3 |
| Manipur | Churachandpur | 3 | 4 |
| Nagaland | Dimapur | 2 | 4 |
| **Total** | | **8** | **11** |

(Also see Supplementary table for profile of interviewed doctors).

Most doctors (6) worked in private facilities, including one at a charitable/mission hospital. Two public doctors in the sample, both from Manipur, also worked at other facilities. All patients interviewed were admitted to private facilities, and most (9) were women. Participants were mostly between 30 and 45, with one male in his twenties and an older female patient who was 52 (Table 2).

Local researchers translated any oral or written communication from interviewees that was not in English. An information sheet detailing the purpose of the study, confidentiality and anonymity clauses, and the implications of their involvement was shared in advance with all participants. Informed written/recorded consent from all participants preceded data collection activities. All interviews were audio-recorded, transcribed into English, and analysed using NVivo V.12 [16].

All data were anonymised by the local team in India, and the authors did not have access to information that could identify individual participants during or after data collection. TE analysed and prepared the survey results using Stata [17]. AV analysed the interviews using NVivo v.12 [16]. Data analysis involved a deductive aspect given the broader focus on health system capacity and user needs as well as a critical inductive dimension wherein themes were extracted from reading gathered data sources. Qualitative data was analysed iteratively, with data gathering and analysis occurring in tandem and broadly guided by the framework analysis approach [18]. In practice, this involved regular debriefs within the local team, which kept a field diary recording personal reflections. Additionally, the UK and India researchers teleconferenced at regular intervals to discuss emerging themes and connections within the data. The local team transcribed interview data, and a single UK-based researcher (AV) then input and coded the data using NVivo v12 [17]. The UK and India lead research leads (AV, MH) also

**Table 2. Profile of patients interviewed.**

| Female, 28–35 | Bongaigaon | Assam |
|---|---|---|
| Female, 30–35 | Bongaigaon | Assam |
| Male, 40–45 | Chirang | Assam |
| Female, 40–45 | Churachandpur | Manipur |
| Female, 52 | Churachandpur | Manipur |
| Female, 35–40 | Churachandpur | Manipur |
| Female, 30–35 | Churachandpur | Manipur |
| Female, 34–40 | Dimapur | Nagaland |
| Female, 44 | Dimapur | Nagaland |
| Male, 22–25 | Dimapur | Nagaland |
| Male, 30–35 | Dimapur | Nagaland |

met to discuss and clarify initial codes and evolving themes. AV wrote the initial draft of the manuscript.

## Facility survey

To conduct the facility census, the team contacted the administrative departments of all listed hospitals in the study districts, requesting access to logbooks and surgical records on condition of anonymity. Additionally, a standardised questionnaire based on the WHO Surgical Assessment Tool (SAT), which is mostly composed of closed questions with yes/no, categorical, and numeric answers.

Panel 2: Facility functionality

The functionality (F) of facilities was captured across six domains:

i) availability of a surgeon 24/7 in the facilities D1[0,1], taking a value of 1 if a surgeon is available 24 hours a day and reduced proportionately for less availability;

ii) availability of basic infrastructure D2[0,1], taking a value of 1 where electricity, water, and oxygen are available 24 hours a day and reduced by 1/3 for each utility not available;

iii) ability to provide blood transfusions D3[0,1], taking a value of 1 if blood-banking and transfusion equipment are available plus ability to cross-match, reduced by 1/3 for each component lacking;

iv) availability of an anaesthetics D4[0,1], taking a value of 1 if an anaesthetist (specialist, doctor, or trained non-doctor) and anaesthetic equipment are available, reduced by ½ for each component lacking;

v) ability to undertake basic lab tests D5[0,1], taking a value of 1 where all essential tests are available (complete blood count, pregnancy, coagulation, urine analysis, and infectious panel of tests) and reduced by 1/5 for each component lacking; and

vi) availability of basic surgical equipment D6[0,1], taking a value of 1 where all 44 items are available, reduced by 1/44 for each item lacking.

Each of these domains was scored and given a weighting of 1/6th in a total index of functionality to deliver surgical services, i.e.,

$$F_j = \frac{1}{6} \sum_{i=1}^{6} D_{ji}$$

Survey data were entered into an appropriate database and then converted to Stata for analysis. Data entry was double-entry (data were entered by two people separately, and then any differences were reconciled) and undertaken by the local research team. The data had no personal identifier, so were anonymous at the individual level. Facility-level identifiers were included to facilitate later linkage to other data sets using geographic coding.

The results below summarise the information from the quantitative and qualitative data to present a composite analysis of surgical capacity and challenges to effective provisioning of surgical care.

## Results

### Overview: Range of surgical procedures and use

The three-month surgical logbooks and aggregate annual reports both provide a similar picture of the main types of surgery provided across the region. Caesarean section (offered by 18/19 facilities providing surgical data), cholecystectomy (12/19), and appendectomy (14/19) constitute 63% of surgery in all facilities and 76% in public facilities (Table 3). A small number of other services, largely provided in non-government facilities, included hysterectomy, laparotomy, and surgical repair of fractures and hernias, making up a further 30% of surgeries. The remaining 6% is made up of more than 20 infrequently provided surgical services. According to the aggregate reports, the bellwether procedures (caesarean section, laparotomy, and fracture fixation) account for 45% of all surgery in all facilities across the region. The survey showed that although open surgery remains the dominant technique, laparoscopic techniques are increasingly common, particularly in private facilities, with around 28% of laparoscopic-amenable surgery carried out in this way.

**Gender and surgery.** The surgical logbooks suggest that women undergoing surgery outnumber men by more than 3 to 1. This ratio falls to 1.4 if caesarean sections and hysterectomy are excluded, highlighting the prominence of these two obstetric procedures in the region. In the facilities surveyed, women outnumber men for most procedures, even those that are typically emergencies, such as an appendectomy. The median age for men undergoing surgery is 39 (interquartile range 30), while for women it is 34 or 36 (IQR 19) when caesarean sections

**Table 3. Type of surgery by provider ownership (based on annual reporting).**

| Procedure | Private | Government | Total | % in private facilities |
|---|---|---|---|---|
| Caesarean section (% of all surgeries) | 2,289 | 1,632 | 3,921 | 58% |
|  | (30%) | (69%) | (39%) | |
| Appendectomy (% of all surgeries) | 1,208 | 65 | 1,273 | 95% |
|  | (16%) | (2%) | (12%) | |
| Cholecystectomy (% of all surgeries) | 970 | 68 | 1,038 | 93% |
|  | (12%) | (2%) | (10%) | |
| Hysterectomy (% of all surgeries) | 869 | 111 | 980 | 89% |
|  | (11%) | (4%) | (9%) | |
| Laparotomies (% of all surgeries) | 568 | 29 | 597 | 95% |
|  | (7%) | (1%) | (6%) | |
| Trauma laparotomy (% of all surgeries) | 291 | 20 | 311 | 94% |
|  | (3%) | (0%) | (3%) | |
| Drainage of septic arthritis (% of all surgeries) | 39 | 250 | 289 | 13% |
|  | (0%) | (10%) | (2%) | |
| Fracture (% of all surgeries) | 247 | - | 247 | 100% |
|  | (3%) | (0%) | (2%) | |
| Hernia (% of all surgeries) | 213 | 34 | 247 | 86% |
|  | (2%) | (1%) | (2%) | |
| Bowel obstruction (% of all surgeries) | 202 | 0 | 202 | 100% |
|  | (2%) | (0%) | (2%) | |
| Ureterorenoscopy (% of all surgeries) | 139 | 36 | 175 | 79% |
|  | (1%) | (1%) | (1%) | |
| Other (% of all surgeries) | 490 | 91 | 581 | 84% |
|  | (6%) | (3%) | (5%) | |
| Total | 7,525 | 2,336 | 9,861 | 76% |

are excluded. There is an even split between surgery carried out under general anaesthesia (47%) and under spinal anaesthesia (46%), with the remainder undertaken using local sedation such as ketamine. If caesarean sections are excluded, more than 70% of surgery is undertaken using general anaesthesia.

**Facility functionality for delivery.**   Based on the 6-domain index of functionality, overall functionality appeared to be a little higher in non-government (median 0.86, IQR 0.19) compared to public (median 0.7, IQR 0.37) facilities. Public facilities were more likely to have facilities for anaesthesia in place, while private facilities appear more likely to provide 24-hour surgical cover, blood-banking, and available surgical equipment (Fig 1).

A positive relationship between facility functionality and surgical volume was evident (Fig 2), reflecting the importance of infrastructure for service provision.

In general, private facilities appear to offer a wider range of procedures, an average of 14 out of the 34 listed in the survey instrument (Table 3). In contrast, public facilities offer 5–6 procedures on average, most commonly: caesarean section, appendectomy, hysterectomy, tubal ligation, laparotomy, and cholecystectomy. Although these results will need to be read with caution as there were significantly fewer (n = 5) public facilities among the universe of surveyed facilities, there appears to be a weak positive association ($R^2$ = 0.41) between the number of different surgical procedures provided by a facility and the functionality of that facility (Fig 2). The private sector dominates the provision of most surgical procedures, although there is almost parity for caesarean sections, and government facilities mostly carry out drainage for septic arthritis.

The assessment suggested that most facilities have adequate infrastructure, the ability to undertake essential laboratory tests, and surgical equipment. Most facilities could provide near 24-hour access to X-ray machines, although access to other scanning technology is much lower, particularly in rural areas (see Fig 3). Interviews with doctors similarly revealed the use of old equipment for surgeries and deficiencies in availability and quality. In some cases, nursing staff were relied on to carry out regular maintenance, such as oiling hand-held equipment, but the nursing staff were not in a position to do this for electrical equipment that needed trained technicians. The following quote from a public facility illustrates this:

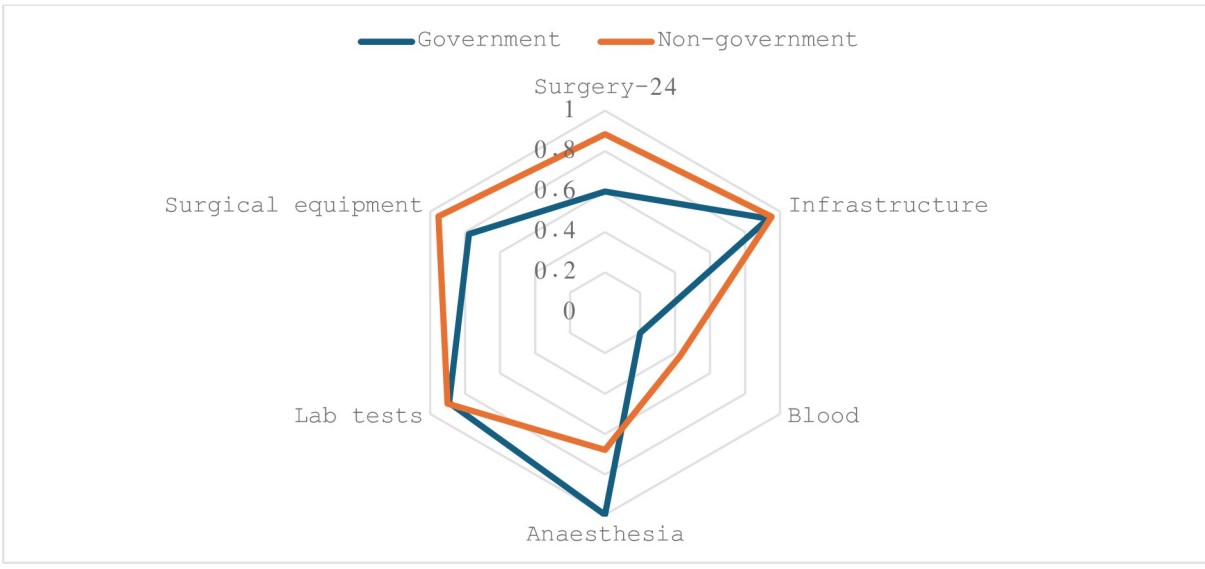

**Fig 1. Domains of facility functionality for public and non-government facilities.**

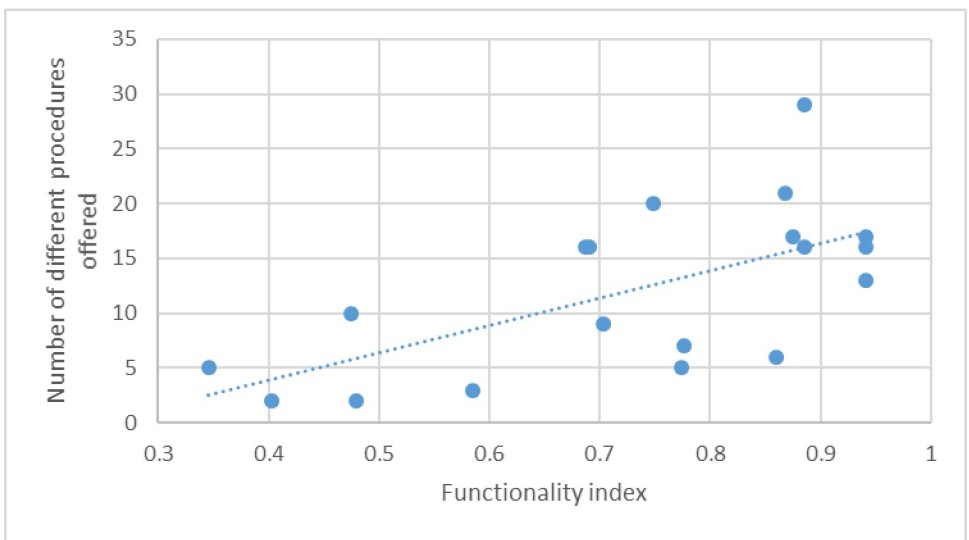

**Fig 2. Number of different surgical procedures and facility functionality.**

*CT scan we don't have. . . we had CT scan but then once it broke down after that nobody is repairing it. The government is also not repairing it. But other than that, like Ultrasound is free of cost, colonoscopy is also free of cost, x-ray also.* (Govt doctor, Manipur, MCPT002MP)

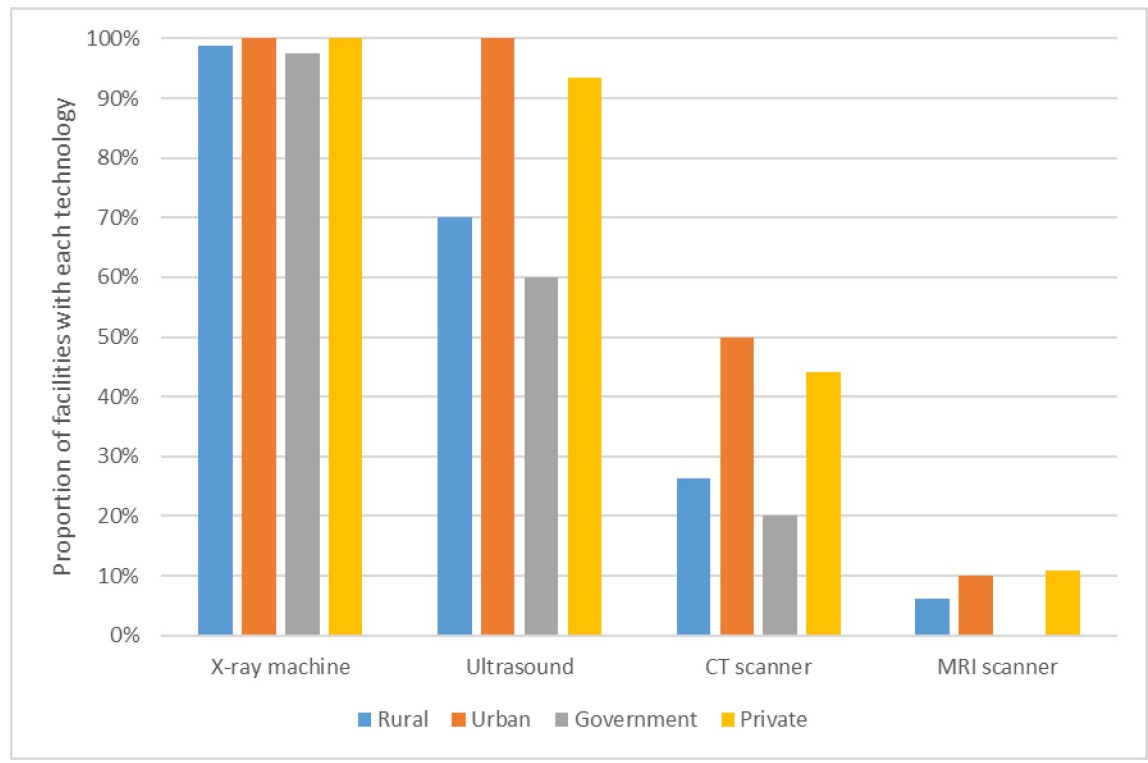

**Fig 3. Availability of scanning technology.**

**Table 4. Mean surgical workload (minimum and maximum per facility in parentheses).**

|  | Surgeons | Anaesthetists | Average number distinct procedures undertaken | Operations per facility | Average operations per different procedure | Operations per surgeon |
|---|---|---|---|---|---|---|
| All facilities | 3.8 (0.5, 9) | 1.75 (0, 4) | 12 (2, 29) | 563 (25, 2924) | 46 (6, 352) | 148 (12, 2924) |
| Private | 4.3 (0.5, 9) | 1.8 (0, 4) | 14 (2, 29) | 595 (25, 2924) | 41 (6, 224) | 137 (12, 2924) |
| Public | 2.2 (1, 6) | 2.2 (0, 3) | 5 (2, 9) | 473 (79, 809) | 84 (11, 352) | 215 (78, 705) |
| Rural | 3 (0.5, 9) | 1.75 (0, 3) | 7 (2, 20) | 342 (25, 809) | 46 (6, 352) | 114 (12, 705) |
| Urban | 4.6 (1, 8) | 1.75 (0, 4) | 16 (3, 29) | 762 (68, 2924) | 46 (12, 224) | 165 (17, 2924) |

## Surgical workloads and capacity

All surveyed facilities had surgeons available 24 hours a day, yet most facilities will often have a single surgeon on shift at any one time. Survey results indicated that the relative paucity of surgeons and anaesthesiologists in some facilities contributes to the huge variation in surgical caseloads. The survey found surgical workloads varying from 25 to almost 3,000 operations per year (Table 4). There are on average 3.8 surgeons per facility, so this equates to 148 operations per surgeon, although some facilities have much higher workloads, particularly in facilities where fewer than one full-time surgeon is employed. The narrower range of procedures offered at government facilities means that while overall surgical loads appear to be higher in non-government facilities, the average number of operations using each procedure is considerably lower in such facilities.

A number of surgeons we interviewed reported feeling overworked, especially because they had to manage outpatients during the day and accommodate surgical patients in the evenings.

> . . . eleven hours is the minimum hours that I spend here. . . Most of the surgeries I have done in the night. . .because I do not get time in the daytime. Whole day (nurses) will be walking around to see the patients. . .the same nurses have to go to surgery. To help me, so they get tired. . . also myself the same thing. (General surgeon, Assam, ACPT002)

The situation is exacerbated by the general unavailability of skilled staff, including assistant doctors and nurses. The lack of anaesthesiologists was a universal problem prominently mentioned in the interviews, with a few also reflecting on the demoralising effect of shortages on their ability to do their work. The lack of dedicated anaesthesiologists, who would come on call, clearly added to the available surgeons' workloads. In some instances, surgeons mentioned multi-tasking, providing anaesthesia themselves, or relying on nurse assistants.

> . . .sometimes we do not get anaesthetist (anaesthesiologists) in time so there we have to do by our self. . .sometimes nurses to give a spinal. . . (General surgeon, Arunachal Pradesh, APBPT002)

Notably, surgeons working on their own, typically in smaller private facilities (<500 patients per annum, 10–20 beds) lamented not having surgical colleagues in-house with whom they could discuss cases and procedures. This peer support was considered particularly critical for more complex cases involving excessive bleeding or suspected tumours. Clinical assistance was valued in such situations both for consultative purposes to guide medical decision-making as well as for practical help in performing more complex procedures.

> . . .anaesthesia is the biggest challenge here and another thing is the investigation facility. Sometimes. . . we can't diagnose. . .because of lack of . . . CT scan, MRI, and endoscopy and

*sometimes we even have shortage of surgeons. . .(ideally) around 2–3 surgeons gather. . .discuss and. . .do the surgery properly. . .(But) we are left. . .alone to do one surgery which is very, very frustrating. . . (General surgeon, Arunachal Pradesh, APBPT002)*

Some made do with existing nursing staff, often ones they had trained on the job. At other times, surgeons would consult more senior colleagues located in bigger cities.

*. . . I'm the only surgeon here and sometimes our medical superintendent, who is general practitioner. . .assist me in doing surgery and that becomes a little helpful for me. Otherwise, I need to take the assistance of our sisters (nurses) who are also very well off now to assist during the surgeries. (General surgeon, Assam, ACPT002)*

Among providers, there was general openness to using less invasive, conventional (using CO2) and gasless laparoscopic techniques, particularly for lower abdominal surgery, appendectomies, and gall bladder procedures. A few reported limited experience during medical camps organised by an external agency or, in the case of public doctors in Manipur, in nearby private facilities where they also practiced. While acknowledging the higher cost for patients, providers generally noted the clinical benefits in terms of less scarring, post-operative pain, and infection risk. While limited experience/expertise combined with specialist equipment shortages constrained more widespread use, further structural problems were observed by a doctor in a district hospital:

*(Government) sanctioning for a laparoscopic set. . .was a problem and then the second thing is the team work. . .Last time . . . the government started giving us some laparoscopic instruments. . .I told the medical superintendent that it's a team work . . .I want the O.T light (electricity) not to be off at least for one month. . . good that we have solar (power), but then solar also sometimes fails and then when the generator person never stays. . .who is also a government servant. And we have examples like we have to wait for about 30 minutes in OT with the gloves because the light went off and we don't have any other backup. (Surgeon, Manipur, MCPT002)*

The above results provide an overall picture of the nature of workload and its effects on existing surgical provision, the implications of which are considered in the discussion.

## Referrals

Across the surveyed facilities, around 6% of patients requiring surgery are referred to other facilities. This level is higher in rural areas than in urban areas and in public facilities than in non-government facilities (Table 5).

Interviews revealed diverging demand and supply-level motivators for referral to other facilities. Facility-level capacities were a principal factor prompting referrals by doctors, especially patients with co-morbidities or chronic diseases. A few doctors, for instance, mentioned having to refer patients when anaesthesiologists were unavailable at short notice. At the facility level, a few doctors indicated diagnostic referrals to nearby private facilities for more advanced investigations like MRI or specialised procedures like endoscopy and laparoscopy. A doctor who held dual jobs at both a public, government-run facility, and a private facility admitted having to routinely refer existing patients to other facilities with functional infrastructure for laparoscopic procedures. Similarly, patients needing specialised surgery or complex cases that required more resources were referred to other facilities.

**Table 5. Surgical referrals.**

| | Surgical referrals | Admissions | Surgeries | Referrals/admission | Referrals /patients needing surgery |
|---|---|---|---|---|---|
| Rural | 57 | 3,382 | 381 | 1.7% | 13% |
| Urban | 42 | 3,062 | 988 | 1.4% | 4% |
| Government | 94 | 3,807 | 387 | 2.5% | 20% |
| Private | 36 | 3,034 | 825 | 1.2% | 4% |
| Total | 48 | 3,197 | 733 | 1.5% | 6% |
| | | | | | |
| Min | 5 | 300 | - | 0% | 0% |
| Max | 200 | 12,063 | 6,000 | 23% | 100% |

*. . .renal failure and . . . cardiac problem. . .so when they have complications, we need to refer and when their conditions is very poor. . . our hospital doesn't have blood bank . . .abdominal traumas suppose someone is having liver injury, we need lots of blood so those kind of patients we cannot keep here. . . (Surgeon, Assam, ABPT002)*

As alluded to in the above quote, the survey revealed considerable variation in the adequacy of blood-banking facilities. Except for one respondent, a patient, who mentioned in-house blood-banking facilities, most patients and surgical providers stated that patients needed to obtain blood from external sources. Family donations or blood banks in other locations were often used. A doctor in a private mission hospital (19 beds, 1 functioning Operation Theatre, approx. 1032 admissions per annum, 2 doctors) mentioned that they would either stabilise those likely to need blood transfusions or rely on their experience before proceeding with a surgery, using patient haemoglobin levels as a yardstick. For such facilities in small towns, other infrastructure priorities and regulatory factors disincentivised investment in blood-banking.

*The licences [for a] blood bank are very difficult to get. We need pathologists. . .dedicated for the blood bank. . .there is lot of norms . . .staffing patterns, the room, the chairs. . . A lot of things are required. . .These will cost lots of money–we cannot keep as a priority a blood bank . . .We ask for the blood but if the haemoglobin is around nine or ten I'll definitely go ahead because getting blood is a [big problem]. (General surgeon, Assam, ACPT002)*

Along with limited equipment, this lack of surgically trained staff was a contributing factor in the less routine use of minimally invasive laparoscopic techniques. Despite the interviewed surgeons' amenability to laparoscopy, some pointed out that they were using it mainly for elective rather than acute cases requiring advanced laparoscopic capability, which were lacking in their facilities. This was because of affordability for patients as well as the surgeons' tendency to avoid potentially more complex cases via laparoscopy, for which they had limited training and experience. Specifically, some surgeons acknowledged the clinical and economic benefits of laparoscopy, including quicker recovery time and less pain, which, given their patient profile of less well-off populations, was likely to be a preference for the patients as well.

Interviews with medical staff indicated that record-keeping varied greatly between facilities. Some facilities digitised records, whereas others kept written registers. In general, public hospital data was regularly sent to government health information management systems.

## Treatment-seeking, cost, and pathways of care

Patients in our sample were generally satisfied with the care received, mainly because they had their ailments addressed. A range of considerations were noted in patients' descriptions of

care experiences—doctor and medical staff manner, patient's prior experience and hence faith in using a certain facility, and for one user in Manipur, a shared dialect with medical staff, which improved communication and comfort level. Good interactions with staff as well as clear communication by medical staff reassured patients, which was important for patient satisfaction. The following respondent, who had their pain symptoms persist and then sought treatment at a private facility, reflected on their earlier experience at a public hospital.

> *I was abruptly discharged. . . what I mean the doctor will come for the round-'she is fine- discharged'. . . so I was a bit disappointed. . . that they discharged me because I do not know what my condition is . . . because the pain is still there. . . I wish the doctors can be more reassuring then they can give us more advice. (Female patient, 40–45 years, private facility, Manipur MCPT001PTE)*

Interviewed patients generally reported self-treating when symptoms, typically abdominal pain, first emerged. Getting medicines from local pharmacies was the preferred option for most patients. Patients in our sample generally preferred consulting medical doctors over traditional practitioners, often because of their faith in formal care as well as symptoms like acute abdominal pain that required emergency or obstetric care. Shorter distances to facilities, the reputation of the doctor/facility, and, for some, lower costs of services were prominent factors affecting patients' initial choice of facility. A few responses also indicated an acceptance of poor roads and traffic conditions, without noting these as factors affecting access to facilities. Convenience and cost emerged as prominent drivers for seeking services from multiple providers, resulting in slightly longer pathways to treatment.

> *I went to the other hospital because it's nearby and my case was an emergency. But I got shifted to this hospital because it is cheaper here. I took doctor's reference to come here. (Female patient, 30–35 years, private facility, ABPT002)*

The facility survey revealed that the cost of essential surgery varies across health facilities but is on average between 5 and 8 times higher in non-government hospitals compared to public hospitals. Average charges in non-government hospitals constitute between 37% (C-section) and 47% (fracture repair) of annual state (averaged across Northeast States) GDP per capita (Table 6). Charges are generally higher in urban facilities compared to rural facilities. The cost of a caesarean section, for example, was reported as being RS 30,000 in urban areas as compared to RS 21,000 at rural facilities.

**Table 6. Facility charges of essential surgery in Indian rupees and US Dollars (minimum and maximum values per facility in parentheses) and as % of GDP per capita across Northeast states).**

| Average (mean) charge | | C-section | Fracture repair | Laparotomy |
|---|---|---|---|---|
| All facilities | Indian Rupees | 27,700 (1,400, 40,000) | 36,385 (1,000, 60,000) | 33,031 (7,000, 50,000) |
| | USD | 386 (19, 557) | 507 (13, 836) | 460 (97, 697) |
| | % GDP per capita | 34% | 44% | 40% |
| Public facilities | Indian Rupees | 3,700 (1,400, 6,000) | 7,000 (7,000, 7,000) | 7,000 (7,000, 7,000) |
| | USD | 51 (19, 83) | 97 (97, 97) | 97 (97, 97) |
| | % GDP per capita | 4% | 8% | 8% |
| Private facilities | Indian Rupees | 30,900 (15,000, 40,000) | 38,833 (1,000, 60,000) | 34,767 (20,000, 50,000) |
| | USD | 430 (209, 557) | 541 (13, 836) | 484 (278, 697) |
| | % GDP per capita | 37% | 47% | 42% |

Note: Rupees are converted to USD at the December 2019 rate using the Oanda currency converter.

## Discussion

The quantitative and qualitative results collectively provided critical insights into surgical care capacities in NE India and current practices.

Abdominal (37%) and obstetric/gynaecological (50%) conditions constituted a large portion of the overall surgeries across public and private facilities in the region. The prominence of caesarean sections was evident, on the positive side signalling the availability of emergency obstetric care, but equally a more portentous trend towards avoidable procedures [19]. This was partly reflected in the higher uptake of surgery among women, which could indicate the magnitude of obstetric health needs in the region while also raising questions about men's health-seeking behaviours.

Facility functionality and being better equipped were enablers for surgical delivery, although other considerations may also explain lower caseloads in some facilities. An accompanying paper, for example, suggests that location of facility and distance are critical factors in determining use of facilities, suggesting lower take-up by users even in cases where surgical resources may be available [12]. Results demonstrated variable workloads for surgical providers—interviewees generally reported excessive workloads, although the facility survey also revealed smaller surgical volumes at certain facilities. This signals a possible association between facilities' location and surgical demand, perhaps those in larger districts receiving a higher number of patients. It may also indicate that those with higher workloads were perhaps more willing to be interviewed. Interviews clearly revealed the pressures that excess workloads placed on the existing personnel, both surgeons and nurses. Scarce peer support for complex cases and a reliance on nursing staff created physical and cognitive demands on surgeons. More broadly speaking, skewed workloads, whether high or low, have critical implications for provider efficiency, patient safety, and care quality. High workloads, for instance, can overburden staff, affecting their motivation and performance and, in some instances, leading to preventable clinical errors and suboptimal treatment [20,21]. Conversely, low surgical volumes at some facilities may limit opportunities for staff to get sufficient practice, leading to adverse patient outcomes. For instance, Mikeljevic and colleagues [22] reported lower survival rates among breast cancer patients treated by surgeons with lower workloads. More broadly, skewed workloads across facilities and regions are indicators of uneven surgical delivery, with a lack of uniformity in care quality and facility performance.

There was some evidence of task-shifting, with non-specialist nursing staff assisting doctors on surgical procedures or surgeons taking on anaesthetic tasks or supervising, as has occurred elsewhere in the US, Western Europe, and Sub-Saharan Africa [23]. In Sierra Leone, task-shifting involves training non-surgeons to perform certain operations rather than nursing staff undertaking dual roles. Mavalankar and Sriram [24] view task-shifting to mid-level providers as a promising strategy given personnel and training shortages for specialised anaesthetic care in South Asia. According to the authors, in addition to better staff retention amongst mid-level providers, effective training and quality assurance measures can ensure the cost-effectiveness and safety of task-shifting approaches in South Asia. Anaesthesia is a vital element of emergency obstetric care for tackling pregnancy-related complications, which are especially prominent in South Asia and account for high rates of maternal mortality.

However, viewed differently, the performance of anaesthesia by surgeons also reflects risk-taking behaviour by hospitals/doctors, which is an important underlying factor affecting health care-seeking behaviours. As a recent study from Assam reports, litigation fears, along with patients with limited means, often lead to reactive and risk-averse measures by secondary-level hospitals in referring patients elsewhere and a lack of responsiveness [25]. In rural, remote settings, an adaptive approach to refashioning standardised practice guidelines and protocols is recommended in order to ensure timely care for patients.

In addition to personnel and infrastructure shortages, shortfalls in in-house blood-banking facilities were evident, adding time and costs to patients' treatment pathways as blood supplies had to often be externally sourced or replaced through family donation. While the shortages in blood supplies globally as well as in India due to a combination of demand and supply level factors are well known, our results also revealed structural disincentives, including tedious licensing norms for in-house storage, discouraging active investment in blood-banking facilities [26,27]. In the absence of adequate personnel (e.g., in pathology, radiology, and anaesthesia), greater internet connectivity may be considered an important enabler to obtain high-quality second opinions and access health-care initiatives being rolled out by the Government of India. Prominently, there was amenability to greater use of minimally invasive laparoscopic techniques, which are routinely used in more developed health systems, given their relative cost, time, and efficiency advantages over open surgery [28]. Readiness for surgical services in Northeast India is affected by the poor availability of supporting consultants. Options for relaxing current government legislation that requires a pathologist to register a blood bank and a radiologist to get PNDT certification (to use ultrasonography) may be considered in such settings where consultants are not available, including the potential for nurse anesthetists. Clearly, however, equipment and training incapacities were acknowledged, thereby signalling the directions for possible policy action.

There are some notable differences between the facilities in urban and rural areas. Urban facilities tend to charge more than rural ones, partly because they are more likely to be private and also perhaps because of a relatively richer population. Urban facilities generally have higher workloads, largely because staffing is higher; workloads (operations) per surgeon are only slightly different.

Survey results revealed high rates of referral for some (particularly public) hospitals to other facilities—often because surgeons or equipment were not available. Interviews with surgeons at private clinics confirmed facility-level infrastructural shortcomings or personal incapacity as factors prompting referrals to external facilities. While case complexity often constitutes legitimate grounds for referral to higher levels of care, our results suggest that avoidable treatment delays were occurring due to a lack of basic diagnostic infrastructure in several mid-range facilities. In our results, the one striking instance of a surgeon transferring public patients to private facilities raises concerns about affordability for patients and equity within the system, especially as we found that for users, cost of care often functioned as a driver for self-referral to cheaper, often public, facilities for procedures. Essential procedures were expensive at non-government facilities, which has implications for affordability and equity of access within the system. More broadly, this example provides further evidence on the factors motivating referral to private facilities [29]. In addition to doctors' personal interest and circumstance as motivators in some contexts, our work shows that doctors' assessment of patients' financial ability and preferences also factor into referral to better-equipped (with investigation technology) or cheaper (public facilities, or electing for more technologically advanced procedures (e.g., laparoscopy).

## Study limitations

Admittedly, the sample of provider and patient interviewees captures only a limited set of views from a few districts. Yet, these accounts provide rich detail on the evidence uncovered through the facility survey, providing a comprehensive overview of surgical delivery in the region. Moreover, while the included patients were generally female and often being treated for abdominal conditions, the interviews nevertheless represent a prominent population and category of surgical need in the country. Difficulties accessing information from the entire

universe of surgical facilities meant that certain categories, namely military and certain missionary facilities, important providers of care in the region, were excluded from the results. The majority of the interview respondents were from the private sector, albeit including those in dual practice. Under ideal circumstances, equivalent representation of the public and private sectors would have provided a more comprehensive overview. However, the sites are broadly indicative of the general capacity for surgical care and the experiences of those privileged enough to be able to access surgical services. In tandem, the quantitative and qualitative assessments provide a useful picture of surgical care in Northeast India.

## Conclusion

Service availability, represented by functioning facilities and adequately trained staff, is vital for surgical access. In low-resource settings, such as Northeast India, informal arrangements for task-shifting and referral function as common strategies for surgical providers to cope with surgical demand. The motivations for referral often stem from facility incapacity in certain specialised areas of care, such as anaesthesia and advanced diagnostics. Such shortages call for structural changes to facilitate procurement and training in priority specialties. Innovations in technology and management practices are needed.

## Recommendations and future research

This study provides directions for future research to consider the factors affecting variable workloads at surgical facilities, more gender-specific patterns of health care seeking behaviours in the region, and mechanisms for improving facilities' responsiveness towards patients.

There is now increased availability of: 1. digital technologies offering affordable and efficient ways to enhance surgical training through remote proctoring; and 2. frugal innovations, such as gasless laparoscopy, both of which may allow patients in rural settings to benefit from minimally invasive procedures. On the organisational side, better coordination of referral among existing facilities could go some way towards ensuring more integrated and efficient services. On staffing, a system-wide approach is needed to ensure a coordinated response to addressing scarcities in human resources and skill mix. Finally, while health system-level levers are vital for enhancing surgical delivery, wider political, economic, and structural change to ensure adequate prioritisation and financial investment will be crucial to address other components of access to care, including economic and geographic aspects.

## Supporting information

**S1 File. Supplementary table: Profile of interviewed doctors.**
(DOCX)

**S2 File. PLOS one statement on inclusivity in global research.**
(DOCX)

## Acknowledgments

Thepfuchanuo Kire, Atshele Venuh, Benrilo Shitiri, Keduokuolie Pienyu, Lungreihingbe B. Riame, Seyievor Yhome.

## Author Contributions

**Conceptualization:** Amrit Virk, Rebecca King, Michael Heneise, Julia Brown, David Jayne, Tim Ensor.

**Data curation:** Michael Heneise, Lanuakum Aier, Catriona Child.

**Formal analysis:** Amrit Virk, Catriona Child, Tim Ensor.

**Funding acquisition:** Julia Brown, David Jayne.

**Investigation:** Amrit Virk, Lanuakum Aier, Catriona Child, Tim Ensor.

**Methodology:** Amrit Virk, Rebecca King, Michael Heneise, Lanuakum Aier, Catriona Child, Tim Ensor.

**Project administration:** Amrit Virk, Rebecca King, Michael Heneise, Catriona Child.

**Supervision:** Amrit Virk, Rebecca King, Michael Heneise, Lanuakum Aier, Catriona Child, Julia Brown, David Jayne, Tim Ensor.

**Writing – original draft:** Amrit Virk, Tim Ensor.

**Writing – review & editing:** Amrit Virk, Rebecca King, Michael Heneise, Lanuakum Aier, Catriona Child, Julia Brown, David Jayne, Tim Ensor.

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
