## [Decision Letter · Decision Letter 0]

31 Jul 2023

PONE-D-23-17522How ready is the health care system in North East India for surgical delivery? A mixed methods study on surgical capacity and needPLOS ONE

Dear Dr. Virk,

Thank you for submitting your manuscript to PLOS ONE. After careful consideration, we feel that it has merit but does not fully meet PLOS ONE’s publication criteria as it currently stands. Therefore, we invite you to submit a revised version of the manuscript that addresses the points raised during the review process.

We look forward to receiving your revised manuscript.

Kind regards,

Lovenish Bains, MS, FNB, FACS, FRCS (Glas), FICS, FIAGES

Academic Editor

PLOS ONE

5. Please include your tables as part of your main manuscript and remove the individual files. Please note that supplementary tables (should remain/ be uploaded) as separate "supporting information" files

Additional Editor Comments:

The manuscript has good information on health system capacity in resource constraint settings in Northeast India; however, on other side it is a classic case of “Helicopter research” when researchers from higher-income or more privileged settings carry out research in resource-poor settings with limited to no involvement of local researchers. There is no Indian Co-author in the Author list.

The world is talking about “Equitable Research Partnerships” and new frameworks being developed to promote greater equity in global collaborations.

The manuscript suggests at many places about the work done by local team “The UK based team worked in close collaboration with regional institutions, including a clinician-led, humanitarian organization providing medical care and surgical training in the region and a team of social scientists from the region.” Despite this, local researchers are not part of the authorship.

The IRB approval is matter of great concern as pointed by few reviewers because the so-called agency is located in New Delhi and is providing Approval for the study in different state. The approval letter says that it is Institutional Review Board but no Institutional attachment. The Indian Council of Medical Research (ICMR), New Delhi, the apex body in India for the formulation, coordination and promotion of biomedical research as per its guidelines provides unique registration number for the IRB and also the process for International Collaborations through HMSC, Health Ministry Screening committee as the local data has moved from the country to another country.

Therefore, the authors must provide the proper approval documents; in the absence of which the manuscript cannot be considered further.

The reviewer comments are attached.

Reviewers' comments:

Reviewer's Responses to Questions

**Comments to the Author**

1. Is the manuscript technically sound, and do the data support the conclusions?

Reviewer #1: Yes

Reviewer #2: Partly

Reviewer #3: Partly

Reviewer #4: Partly

Reviewer #5: Yes

Reviewer #6: Yes

Reviewer #7: No

Reviewer #8: Yes

2. Has the statistical analysis been performed appropriately and rigorously? 

Reviewer #1: Yes

Reviewer #2: No

Reviewer #3: N/A

Reviewer #4: I Don't Know

Reviewer #5: N/A

Reviewer #6: Yes

Reviewer #7: N/A

Reviewer #8: I Don't Know

3. Have the authors made all data underlying the findings in their manuscript fully available?

Reviewer #1: Yes

Reviewer #2: No

Reviewer #3: Yes

Reviewer #4: No

Reviewer #5: No

Reviewer #6: Yes

Reviewer #7: No

Reviewer #8: Yes

4. Is the manuscript presented in an intelligible fashion and written in standard English?

Reviewer #1: Yes

Reviewer #2: Yes

Reviewer #3: Yes

Reviewer #4: Yes

Reviewer #5: Yes

Reviewer #6: Yes

Reviewer #7: Yes

Reviewer #8: Yes

5. Review Comments to the Author

Reviewer #1: I enjoyed reviewing this paper, because:

1. The research question is well defined.

2. The methodology is appropriate.

3. Statistical analyses in result section are appropriate and figures and tables highlight the trends well.

4. In discussion authors have done well to discuss all relevant points and the limitations of the study are acknowledged.

My two major concerns are:

1. There is very little new in this manuscript; the deficiencies in rural remote areas are well known, as is the solution. However, lot of hard work has been put in this survey; hence it gets the nod from me.

2. A bigger concern is with the ‘Helicopter’ style of research in which there are no local authors, although all the data is ‘local’.

The call on the ethical suitability of such a paper being published in the journal has to be taken by the editorial team.

Reviewer #2: The selected topic is relevant to policymakers and there is paucity of literature on it - therefore the study is justified.

Considering a global audience for this topic, there should be a map as well as introduction to northeast India with a short description of its uniqueness (geography, ethnicity, history, factors such as insurgency etc.)

There are many factors that affect health-seeking behaviour, especially in remote rural locations in India. Risk taking behavior by hospitals/doctors (such as performance of anaesthesia by surgeons or referral to higher centers for trivial reasons) have been mentioned but these should be explored in detail in the design of the study. This is an article analysing some of these factors from northeast India: https://casereports.bmj.com/content/15/7/e248221 - this is a case report that analyses global health factors, there should be other studies as well.

The sampled data is too small and not representative of the region. Data has been collected from only 4 districts/states, 8 doctors and 11 patients. There are several large hospitals both in the government and private sector which manage large volumes of patients. These should have been included.

With a high 'out of pocket' expenditure in healthcare, Indian costs should be analysed from the patient's point of view. It will be good to visit large, medium and small government and private hospitals to make a detailed analysis of how much the patient has paid towards service - those admitted to government hospitals also spend significant amounts to purchase drugs/supplies or non-available diagnostics. They also face significant non-hospital expenditure such as travel, loss of wages, food etc.

Indian states have adopted their versions of the Clinical Establishment Act which enumerates a number of conditions that must be met for registration. Laws applicable to blood banking and ultrasound are unique to India and a major barrier in northeast India due to the paucity of pathologists and radiologists. These are an important barrier to service provision (especially in remote rural areas of the region).

Besides electricity, water and oxygen, internet access could be considered as one of the critical requirements. It enables high quality second opinions and the ability to log in to government and private portals for the many health care initiatives that are being rolled out by the Government of India. Besides technical and cost issues, internet access in many parts of northeast India are cut by the government for prolonged periods of time to tackle insurgency and other security issues.

Distance to nearest center offering the service for which the patient has been transferred would be an important factor to measure.

Surgical workload should mention how many surgeries among how many surgeons over what time period. Workloads ranging from 25 to 3000 surgeries appears to refer to institutional numbers per year.

Considering a global readership, costs should be converted from INR to USD.

A very similar study (Ensor T. et.al.) with some of the same authors explores obstetric and gynaecologic surgeries in the same region. It is a much more comprehensive and better designed study. This study could borrow design and analysis features from the earlier study but include other general surgeries or omit the obstetric and gynaecological surgeries to derive greater value.

India offers a comprehensive health insurance scheme. The RSBY scheme has been mentioned in the above study but this has been replaced by the Ayushman Bharat scheme which is modified by different states. Access to this scheme with its barriers and enablers, reimbursement time, opt-in/opt-out rates etc. would also affect surgical service accessibility.

Fracture repair and laparotomy are very broad descriptions. It could be better to be more specific - appendectomy, laparoscopic cholecystectomy, open reduction and internal fixation of a trochanteric fracture, open mesh hernioplasty etc. would be good examples.

The list of references has several duplications (same article mentioned twice).

Reviewer #3: This study entitled ‘How ready is the health care system in North East India for surgical delivery? A mixed methods study on surgical capacity and need’ addresses the scarcity and unequal access to surgical services in Northeast India, an area with a high surgical need but sub-par delivery rates. The study uses mixed methods, including facility-based census and interviews with surgeons and patients, the research provides valuable sub-regional data on surgical provisioning and need. The authors have shown commendable efforts in organizing the manuscript in a comprehensible manner. Still, certain suggestions require attention and should be rectified by the authors.

1. Page No. 3, What does author means about (Box 1), is it a typo? As the Box1 is cited in the manuscript but mentioned in the manuscript itself? The author should thoroughly review and ensure consistency throughout the manuscript.

2. Page No. 5, The authors have discussed the interviews conducted in Nagaland, Assam, and Manipur, but they have not mentioned anything about the interviews in Arunachal Pradesh, even though this state was included as one of the four selected states for the current study. Additionally, authors should justify about the exclusion of interview in Arunachal Pradesh.

3. Authors have cited Panel 1 at two distinct places. Cite at the relevant place only.

4. The authors are advised to verify the citation given on Page No. 5 under the section 'Site selection' for SRQR, as it is challenging for the reviewer to evaluate the specific details of the methods used in this study.

5. Is the data provided in Table 3 applicable to all four states? If so, it would be better to present the data separately for each state. By doing this, it will enable a state-wise comparison and facilitate better interpretation of the healthcare facility across the states.

6. The total referrals calculation in Table 5 seems to be incorrect. The authors are strongly encouraged to thoroughly review all the calculations for accuracy. Furthermore, it is suggested that the authors should rectify the percentage calculation in the Referrals/admission column, as it may be confusing. For example, the reported 5.5% for the Rural population in the Referrals/admission column requires clarification.

7. Also, it would be better to mention about the number of the patient out of total Admissions that were excluded from the referrals and that were underwent surgeries.

Reviewer #4: Evaluation of data collected by 7 member team of local researchers in India has been done by authors based in the United Kingdom without anyone of them having any personal interaction with the study population.There was no uniformity in the doctors and patients interviewed, majority were from private sector.The sector of population , facilities and medical personel covered though mentioned as being uniform is not so. More so because data collected has been transmitted by a intermediary.

Reviewer #5: Summary: The manuscript presents a mixed methods study investigating various aspects of surgical care service delivery & impact in Northeast India. I congratulate the authors on conducting a valuable study in a geographic region that requires more attention in global surgery. The study brings out rich data. I don't have any major concerns/issues. However, I believe that the manuscript could benefit from the following suggestions. All the best to the authors!

Comments:

Citations in multiple places seem misplaced. E.g., in the Introduction section, “Yet, a staggering 5 billion people – approximately 5 of every 7 people globally – lack access to safe, timely, and affordable surgical care (Jamison et al 2013; Debas et al 2015)” should cite Alkire et al. 2015 (https://pubmed.ncbi.nlm.nih.gov/25926087/) that gave the 5 billion number for access. Similarly, the statement - “The figures reveal vast geographical disparities, 295 in LMICs against 23,000 surgeries per 100,000 population in high-income countries, driven by shortages of medical staff, poor access to healthcare services, and weak record keeping (Bhandarkar et al 2021)” also does not seem to have an appropriate citation. Bhandarkar et al is about one tertiary hospital in India and could not have given numbers on average surgical volumes in HICs vs. LMICs. Where are the 295 and 23000 numbers from? I would suggest that authors cite the source.

Discussion and Conclusion in the abstract should start with an uppercase letter - “Skewed workloads …”

In the Introduction, “This region is also distinct in Eurasia, characterized by ethnolinguistic diversity, indicating that many vernacular health systems interact (Post et al 2022)” - I am unsure if the term “Eurasia” is appropriate here.

In Methods, the authors point to Box 1 which has the survey tool. But I could not find the box in the manuscript.

Similar to the profiles of patients interviewed in Table 2, information on doctors would be important - degree qualification (MBBS vs. MS/MD), subspeciality, years of practicing experience, gender, etc.

Under the Methods - Facility Survey section, the authors mention - “Data entry was double-entry and undertaken by the local research team.” What do authors mean by double entry here? A clarification would be helpful for the readers.

Table 3 has percentages added across rows. However, it might be also interesting to have the percentages across columns for depicting public-private comparisons. Maybe another table or a bar chart could do that.

The current Table 3 can be cleaned up to have just three columns for the variables instead of six. Add the percentages in parentheses in the same cell. E.g., 2289 (30%).

I would recommend that authors use "public" & "private" consistently throughout the manuscript text, tables, figures, and panels to improve readability.

Figure 2 should mention a correlation value to quantify the association.

Additionally, the caption needs correction - “Figure 2: Number of different surgical procedures and facility functionality”

Table 4 should specify what values are being noted for Surgeons, Anesthetists, and other variables. Are those mean values or median? Otherwise, 3.8 surgeons - a fractional number - is confusing.

Table 4 can be made concise by having min-max values in the same cell as that of the summary value. E.g., 3.8 (0.5, 9.0). That will reduce the number of rows by a third & make the table more readable.

Table 6 can be cleaned up in the same way as Table 4. E.g., Average charges (min, max) = 27,700 (1400, 40,000).

Also, ideally, it would be better to transpose the table. I.e., switch columns and rows after reducing the cells in an above-mentioned manner. That way, it'll have 3 rows as procedures & 2 columns as average (min, max) and %gdp per capita values.

Is the GDP here Indian GDP or the average GDP of NE states? Please specify.

Figure 3 requires an axis label for the Y axis.

Also, since these are % values, they can range only from 0 to 100%. Please limit the Y-axis accordingly.

In the first paragraph of the Discussion, the authors mention - "... and the readiness for adoption of new surgical innovation.". However, the aims and study data do not support that. Readiness would depend on some demonstration of clinicians' acceptability toward new technology. I would recommend rephrasing it.

In the second paragraph of the Discussion, the authors remark - "Abdominal conditions constituted a large portion of the overall…". However, it's not clear in the manuscript, especially in the Methods, what all conditions were included in the facility surveys & the classification was conducted. The Introduction also primarily focuses on Bellwether procedures that do not include abdominal conditions.

In the third paragraph of the Discussion, the authors write - "... interviewees generally reporting excessive workloads although the facility survey also revealed smaller surgical volumes at certain facilities." This is a truly interesting finding that points to some disconnect in the data on workload & perception of the workload. Could the authors please add a few competing hypotheses/rationale for why this might be happening?

The fourth paragraph of the Discussion - on task shifting - is quite well described. A couple of pointers that the authors can choose to include. First, is there a way to measure task-shifting and -sharing? Can authors perhaps add a metric in their indicators to capture that? If so, that would be a valuable addition that will enhance the study. Second, can the authors discuss the regulatory context? Are nurses allowed to help in or provide anesthesia? Do the findings of this study support changes in regulations that would permit nurses and non-specialist doctors to take up better roles in surgical care? Comments on such areas would make the discussion better.

I would recommend that authors add a paragraph in the Discussion that clearly highlights/summarizes the rural-urban & public-private differences for various parameters. The sixth paragraph does this partially but only in the context of referrals. The comparisons form an important takeaway of the valuable data cited in the manuscript that should be reiterated for the readers.

In Study Limitations, authors call upon their findings as "broadly representative". I would recommend using a more appropriate alternative as collecting representative data is not typically supported by the design of a mixed methods study.

Authors should also consider changing any other such mentions of representativeness throughout the manuscript.

The last six references seem to have been numbered (unlike others) in the bibliography which may not be the correct numbers.

I would recommend proofreading the manuscript once before resubmission.

Points of appreciation:

The manuscript flow is quite good in most places.

The figures are easy to comprehend and present the main takeaways clearly.

The facility functionality indicators are a great addition and I hope other researchers apply them in their work going ahead.

Reviewer #6: Thank you for giving me the opportunity to review this excellent manuscript. The study design and approach is sound, giving a reliable view of the situation in the area under study. I have the following minor comments that may help improve the manuscript:

From the introduction: "Three procedures, caesarean section, fracture repair, and laparotomy together known as the Bellwether Procedures can treat a vast majority of surgical problems; ensuring they are available 24/7 at all first level hospitals a key strategy to expand surgical access (O’Neill 2016)". I am sure the authors are fully aware that more than directly treating surgical problems directly, the capacity to safely conduct these procedures translates into the ability of the surgical system to take care of the majority of first level procedures (for example, if a laparotomy can be performed, then abscess drainage, circumcision, chest tube placement, etc can also be conducted safely). The Bellwethers thus act as “proxy” or indicator surgical procedures. This is an important nuance that needs to be laid out clearly for the reader who may be new to this space.

Is there any way to better chart the sub-regional area studies? Geographic area? Total population of the states under review as a percentage of the overall population of the country? Some general information that would better define the scope of the study, and in turn help the reader to understand its contribution to the overall health scenario of this very large country.

Patients from private facilities; 9 out of 11 were women. Were any steps taken at the study design level to ensure a better gender representation? That may have added valuable insights.

Consider using the term anesthesiologists instead of anesthetists or anesthetics (not sure what the latter term means).

Page 12: "Although these results will need to be read with caution as there were significantly fewer (n=5) public facilities among the universe of surveyed facilities. There appears to be a positive association between the number of different surgical procedures provided by a facility and the functionality of that facility (Figure 2)." Comma rather than a full stop between the two sentences might read better.

Table 6: USD conversion might make this easier for the international reader, especially given changing currency conversion rates. Perhaps both INR and USD figures can be included in sub-columns, using a standard method for conversion (mid point for the study period, or mean conversion rate for the months that the data collection was conducted). Also, can the table be divided by subheadings into overall, government and private sections? That would make it easier for the reader to see at a glance.

Is there any way to estimate the number of surgeons/surgeries from the data collected in this study or from literature from this area? I would assume probably not given the study design, but since it is part of a bigger study, I thought it might be worth asking if this data is available or possible to extrapolate.

Reviewer #7: Thank you for the opportunity to review this paper.

I only have one comment:

this paper does not include evidence of legitimate ethical approval in India. The authors stated that the approval was obtained from Sigma Research and Consulting. I have taken the liberty to look at their website and this is a private consulting company, and no information about whether they are a recognised body to grant ethical approvals is included. This disqualifies this paper from the possibility of being published in a reputable journal because in-country approval is a prerequisite to collecting data in India. Sigma Research and Consulting is not listed as an accredited REC in India.

Before this paper can be reviewed please provide evidence that the study has been carried out in accordance with good practice in research.

Reviewer #8: Thank you for the opportunity to review this very relevant context specific paper on the surgical capacity and barriers to accessing surgical care in North east India. This paper shows the strength of collaboration across two countries to improve research on this very important topic. The paper is done as a mixed methods study. The Quantitative aspect which was conducted as interviews helps to get the practical unmet needs of the community and the surgical workforce and this helps to identify the solutions to solve them as well.

Introduction: presented well

Methodology: For the facility survey was a standardized tool used? If the tool was not a standardized one, how was it validated? The 6 domains have been explained well. In one of the domains which is regarding the availability of basic surgical equipment 44 items have been ,mentioned. It would be useful to have it listed as an appendix or in the methodology.

Results: Presented adequately. In Table 3, the list of surgical procedures done at these centers have been listed. It would be useful to classify the procedures as essential, non essential and emergency as per the Disease Control Priority list (DCP 3). That would enable the reader to understand the nature of the procedures that the center is equipped to handle.

Discussion: presented well. There is a mention of how the paper shows the readiness for adoption of new surgical innovations. But there is no variable highlighted in the methodology or the results to substantiate the same. Was a related question asked in the interviews that were conducted? If that was done, it will be useful to mention regarding the related question asked and the response got in the results.

Limitations: Presented well. The sampling could have been done better to have a near equal representation from the public and the private sector to eliminate the bias. This can be mentioned in the limitation as the majority of the responses were from the Private sector.

Conclusion: The recommendations and the solution presented needs to be separated under another heading. As they discuss points that have not been studied in this particular paper.

References: Adequate

6. PLOS authors have the option to publish the peer review history of their article (what does this mean?). If published, this will include your full peer review and any attached files.

Reviewer #1: **Yes: **Professor Dhananjaya Sharma

Reviewer #2: **Yes: **Vijay Anand Ismavel

Reviewer #3: No

Reviewer #4: No

Reviewer #5: **Yes: **Siddhesh Zadey

Reviewer #6: **Yes: **Lubba Samad

Reviewer #7: No

Reviewer #8: No

---

## [Author Response · Author response to Decision Letter 0]

11 Oct 2023

We fully appreciate and take on board the fact that the authors list can represent those in roles not directly concerned with manuscript preparation and the importance of including local authors. We would like to mention that the lead author (AV) is Indian and has lived and worked in the country for over 20 years, as has MH who led the work on the ground as founder and Director of the local institute in Kohima. However, in cognisance of the importance of naming data collectors in manuscripts, and mindful of ICMJE criteria for authorship, we have additionally included the two local researchers (LA and CC) who led the two arms of the study, and made inputs into the analysis. In compliance with ICJME guidance local researchers (MH, LA, CC) have also read and approved the final manuscript. Kindly note that RK and MH are joint second authors as also noted in the revised manuscript.

---

## [Decision Letter · Decision Letter 1]

13 Nov 2023

PONE-D-23-17522R1How ready is the health care system in North East India for surgical delivery? A mixed methods study on surgical capacity and need.PLOS ONE

Dear Dr. Virk,

Thank you for submitting your manuscript to PLOS ONE. After careful consideration, we feel that it has merit but does not fully meet PLOS ONE’s publication criteria as it currently stands. Therefore, we invite you to submit a revised version of the manuscript that addresses the points raised during the review process.

We look forward to receiving your revised manuscript.

Kind regards,

Lovenish Bains, MS, FNB, FACS, FRCS (Glas), FICS, FIAGES

Academic Editor

PLOS ONE

Additional Editor Comments:

The authors have done significant improvements in the manuscript as per the reviewer's suggestions. The reviewers have accepted majority of the corrections. Few concerns remaining are listed.

The IRB approval is still a matter of great concern. Authors are required to provide proper approval documents and clarifications in this regard.

In the country (India), there is no such body which provides approval of any research in other state while itself being in another state. Local IRB (state/institution) approval will be required.

The Indian Council of Medical Research (ICMR), New Delhi, the apex body in India for the formulation, coordination and promotion of biomedical research as per its guidelines provides unique registration number for the IRB and also the process for International Collaborations through HMSC, Health Ministry Screening committee as the local data has moved from the country to another country.

Do provide the unique registration number for the mentioned IRB.

It must be noted that, in the absence of proper ethical or IRB approvals, the manuscript will not be processed further.

Reviewers' comments:

Reviewer's Responses to Questions

**Comments to the Author**

1. If the authors have adequately addressed your comments raised in a previous round of review and you feel that this manuscript is now acceptable for publication, you may indicate that here to bypass the “Comments to the Author” section, enter your conflict of interest statement in the “Confidential to Editor” section, and submit your "Accept" recommendation.

Reviewer #1: (No Response)

Reviewer #2: All comments have been addressed

Reviewer #3: All comments have been addressed

Reviewer #5: All comments have been addressed

Reviewer #6: All comments have been addressed

Reviewer #8: All comments have been addressed

2. Is the manuscript technically sound, and do the data support the conclusions?

Reviewer #1: Yes

Reviewer #2: Partly

Reviewer #3: Partly

Reviewer #5: Yes

Reviewer #6: Yes

Reviewer #8: Yes

3. Has the statistical analysis been performed appropriately and rigorously? 

Reviewer #1: N/A

Reviewer #2: I Don't Know

Reviewer #3: Yes

Reviewer #5: N/A

Reviewer #6: Yes

Reviewer #8: N/A

4. Have the authors made all data underlying the findings in their manuscript fully available?

Reviewer #1: Yes

Reviewer #2: No

Reviewer #3: Yes

Reviewer #5: No

Reviewer #6: Yes

Reviewer #8: Yes

5. Is the manuscript presented in an intelligible fashion and written in standard English?

Reviewer #1: Yes

Reviewer #2: Yes

Reviewer #3: Yes

Reviewer #5: Yes

Reviewer #6: Yes

Reviewer #8: Yes

6. Review Comments to the Author

Reviewer #1: Authors have addressed one of my comments by including two local authors but the crucial matter of non-availability of "local" ethical committee clearance remains and the final call on this very important issue has to be taken by the Journal

Reviewer #2: 1. The sample size is too small to reflect the reality of surgical services in northeast India. This has not changed in the revised manuscript. It is suggested that more centers with more interviews with surgeons and patients be conducted to make this study more representative. Interviewing several surgeons from the same center is unlikely to add information. Interviews can be done through phone calls or videoconferencing without physically visiting the centers.

2. There are several references to gasless laparoscopy. How many surgeons interviewed perform gasless laparoscopy? What is their experience (number of surgeries performed and outcome)? In most settings, if a surgical team is able to do laparoscopic surgery, getting carbon dioxide gas is not a problem - it is cheap and easy to obtain in most places where oxygen is obtained. Pneumoperitoneum increases working space, visibility and reduces risk.

3. There is a mention of dual-position surgeons who work in both government and private facilities. The absence of surgeons full-time in government facilities and referral of patients to private facilities (where often the same surgeons operate) affects patient access to affordable surgical care. Suggested solutions to the issues raised have not been made.

4. Justified risk-taking should be elaborated. When is it acceptable to take risks in remote settings? If an anesthesiologist is not available or adequate blood is not available, is it justified to take risks or refer to a higher center many hours by road at additional expense?

5. Readiness for surgical services in northeast India is affected by the poor availability of supporting consultants (compared to other parts of India) - differences between northeast India and the rest of the country has not been described. Government legislation that requires a pathologist to register a blood bank and a radiologist to get PNDT certification (to use ultrasonography) should be relaxed/exempted in settings where such consultants are not available. Nurse anesthetists work in many resource-poor settings across the world - this could be considered.

6. Northeast India has many locations where road travel is very difficult or not possible during the monsoon. Patients are evacuated at reasonable cost by helicopter if they can be brought to the nearest helipad (especially in states like Arunachal Pradesh). However, this may not be possible when the weather is bad or when there is an unscheduled emergency. This factor affects readiness to access surgical services and should be described.

7. Surgeons (and other consultants) do not want to work in remote areas because of perceived poor personal and professional life. How many allotted posts for surgeons remain vacant in the region? When a surgeon joins a remote location, how many days is he present on duty? (it is likely that many surgeons do not stay in their remote locations).

8. Time (and life) is lost in transit. Readiness can be assessed by how often a patient who should have been managed effectively is referred 'to a higher center'. What are the reasons for referral?

9. Northeast India has large cities with large medical colleges and private hospitals. To generate data that represents the entire northeast, these should also be included in the study.

10. For the greatest impact on surgical care delivery, what are the recommendations for prudent investment in facility development, human resources and financial support?

Reviewer #3: The authors have responded to all the comments, and the manuscript is now ready for acceptance and publication.

Reviewer #5: Authors have satisfactorily reworked the manuscript. I just have one comment - in the map added to the revised version, J&K and Ladakh seemed to have cut-off. I am unsure if it is an issue with the PDF copy that was generated or something else. But it needs to corrected.

Reviewer #6: Within the remit of the study, the authors have made a heroic effort to address the detailed comments of 8 reviewers!

Reviewer #8: Thank you for the detailed responses to the comments and incorporating the suggested comments. It reads better now.

For the abstract results section, the quantitative results could be incorporated to support the statements mentioned. That would give a better perspective to the reader.

Kindly run the document through a spell check, there are places in the Introduction where 2015 is depicted as 2105,etc.

7. PLOS authors have the option to publish the peer review history of their article (what does this mean?). If published, this will include your full peer review and any attached files.

Reviewer #1: **Yes: **Professor Dhananjaya Sharma

Reviewer #2: **Yes: **Vijay Anand Ismavel

Reviewer #3: No

Reviewer #5: **Yes: **Siddhesh Zadey

Reviewer #6: **Yes: **Lubna Samad

Reviewer #8: No

---

## [Author Response · Author response to Decision Letter 1]

16 Dec 2023

Response to Reviewers-16 December 2023

Subject: PLOS ONE Decision: Revision required [PONE-D-23-17522R1]

How ready is the health care system in North East India for surgical delivery? A mixed methods study on surgical capacity and need.

Dear Dr. Bains, 

Many thanks to you and the reviewers for the careful consideration of our revised manuscript, and for your valuable comments.

As requested, we have appended the following documents,

- Response to reviewers (below)

- A marked-up copy of the manuscript that highlights changes made to the original version, labeled 'Revised Manuscript with Track Changes'.

- An unmarked version of the revised paper without tracked changes, labeled 'Manuscript'.

We believe we have now addressed the reviewers’ comments, and very much look forward to a positive review of this revised version.

Kind regards,

Amrit Virk (on behalf of all authors)

Additional Editor Comments:

The authors have done significant improvements in the manuscript as per the reviewer's suggestions. The reviewers have accepted majority of the corrections. Few concerns remaining are listed.

1. IRB approval : Authors are required to provide proper approval documents and clarifications in this regard. 

2. Do provide the unique registration number for the mentioned IRB.

Response: We have attached the (i) Sigma approval letter and invoice, both listing the IRB Number 10077/IRB/D/18-19, as well as (ii) a Word document with research where SIGMA approval was used. 

SIGMA are registered as an IRB with the Office for Human Research Protections (OHRP) which is a US database of recognised IRBs – see https://ohrp.cit.nih.gov/search/irbsearch.aspx?styp=bsc IORG Number IORG0008260. Importantly, the OHRP database mentioned above is the standard used by the University of Leeds (UK) ethics committee as well, in order to ensure compliance and determine whether in-country ethics boards are recognised.

Our team recognizes the criticality of adhering to ethical standards and addressing IRB approval concerns. And we respect the attention this issue has received from reviewers. 

We partnered with the Highland Institute, known for its independent research in indigenous communities and fluency in 16 local languages. Their previous collaboration with the Nagaland Health Project (Ministry of Health) involved both extensive surveys and in-depth ethnographic research, demonstrating significant breadth and depth, but also anthropological expertise. SIGMA IRB's national scope and experience in various Indian states made them a fitting choice for our study's needs. Our study, focusing on sociocultural and behavioural aspects, diverged from the typical biomedical focus of ICMR guidelines, which follow a framework centred on clinical trials, biological samples, and genetic data. Our international collaboration centred on sociocultural research, did not necessitate HMSC approval. All raw data was securely housed at the Highland Institute and shared internationally only after thorough transcription, translation from as many as seven languages, analysis, and anonymization, upholding strict ethical standards. Our method, reflective of medical anthropology, adheres to research ethics, sharing only processed data internationally, as is common in global research collaborations.

Reviewer's Responses to Questions

Comments to the Author

6. Review Comments to the Author

Reviewer #1: Authors have addressed one of my comments by including two local authors but the crucial matter of non-availability of "local" ethical committee clearance remains and the final call on this very important issue has to be taken by the Journal

Response: Thank you. Our team recognizes the criticality of adhering to ethical standards and addressing IRB approval concerns. And we respect the attention this issue has received from reviewers. 

We have attached the (i) Sigma approval letter and invoice, both listing the IRB Number 10077/IRB/D/18-19, as well as (ii) a Word document with research where SIGMA approval was used. 

SIGMA are registered as an IRB with the Office for Human Research Protections (OHRP) which is a US database of recognised IRBs – see https://ohrp.cit.nih.gov/search/irbsearch.aspx?styp=bsc IORG Number IORG0008260. Importantly, the OHRP database mentioned above is the standard used by the University of Leeds (UK) ethics committee as well, in order to ensure compliance and determine whether in-country ethics boards are recognised.

We partnered with the Highland Institute, known for its independent research in indigenous communities and fluency in 16 local languages. Their previous collaboration with the Nagaland Health Project (Ministry of Health) involved both extensive surveys and in-depth ethnographic research, demonstrating significant breadth and depth, but also anthropological expertise. SIGMA IRB's national scope and experience in various Indian states made them a fitting choice for our study's needs. Our study, focusing on sociocultural and behavioural aspects, diverged from the typical biomedical focus of ICMR guidelines, which follow a framework centred on clinical trials, biological samples, and genetic data. Our international collaboration centred on sociocultural research, did not necessitate HMSC approval. All raw data was securely housed at the Highland Institute and shared internationally only after thorough transcription, translation from as many as seven languages, analysis, and anonymization, upholding strict ethical standards. Our method, reflective of medical anthropology, adheres to research ethics, sharing only processed data internationally, as is common in global research collaborations.

Reviewer #2: 1. The sample size is too small to reflect the reality of surgical services in northeast India. This has not changed in the revised manuscript. It is suggested that more centers with more interviews with surgeons and patients be conducted to make this study more representative. Interviewing several surgeons from the same center is unlikely to add information. Interviews can be done through phone calls or videoconferencing without physically visiting the centers.

Response: Thank you for this suggestion, which we will consider in case of future iterations of this work. The purpose of qualitative interviews in general, and for this study in particular, was to explore the issue of surgical services in greater depth, and complement the information emanating from the quantitative component. The aim is not to generalise or claim to be representative; instead, we aimed to illuminate and provide deeper insights into the challenges and opportunities for improved surgical provisioning. 

2. There are several references to gasless laparoscopy. How many surgeons interviewed perform gasless laparoscopy? What is their experience (number of surgeries performed and outcome)? In most settings, if a surgical team is able to do laparoscopic surgery, getting carbon dioxide gas is not a problem - it is cheap and easy to obtain in most places where oxygen is obtained. Pneumoperitoneum increases working space, visibility and reduces risk.

Response: Thank you, this clinical information is duly noted. A few surgeons in the sample had received training in the technique of gasless laparoscopy, but since this is not a mainstream service, we explored amenability to this technique on the part of surgeons, but were not aiming to study the association between the number of surgeries performed and the outcome. Notably, there was no specific comparison made regarding the number of surgeries conducted using either Gasless procedures and traditional laparoscopic procedures. 

3. There is a mention of dual-position surgeons who work in both government and private facilities. The absence of surgeons full-time in government facilities and referral of patients to private facilities (where often the same surgeons operate) affects patient access to affordable surgical care. Suggested solutions to the issues raised have not been made.

Response: Thank you for this very pertinent point. Given this was not a widely observed pattern in the sample of interviewed providers, we deliberately focused on more prominent and evident discussion points related to the main aim of understanding barriers and opportunities for improving surgical care. 

4. Justified risk-taking should be elaborated. When is it acceptable to take risks in remote settings? If an anesthesiologist is not available or adequate blood is not available, is it justified to take risks or refer to a higher center many hours by road at additional expense?

Response: Thank you. As per the suggestion from the reviewer in the earlier iteration, we referenced to the related study that discussed that adaptative approach, (page 30, lines 469-475), providing a reference point for readers interested in deeper engagement with the issue. 

5. Readiness for surgical services in northeast India is affected by the poor availability of supporting consultants (compared to other parts of India) - differences between northeast India and the rest of the country has not been described. Government legislation that requires a pathologist to register a blood bank and a radiologist to get PNDT certification (to use ultrasonography) should be relaxed/exempted in settings where such consultants are not available. Nurse anesthetists work in many resource-poor settings across the world - this could be considered.

Response: Thank you for this. We have inserted a few sentences conveying this on page 31, lines 489-493.

6. Northeast India has many locations where road travel is very difficult or not possible during the monsoon. Patients are evacuated at reasonable cost by helicopter if they can be brought to the nearest helipad (especially in states like Arunachal Pradesh). However, this may not be possible when the weather is bad or when there is an unscheduled emergency. This factor affects readiness to access surgical services and should be described.

Response: Thank you. We have inserted this information on page 4, lines 66-70.

7. Surgeons (and other consultants) do not want to work in remote areas because of perceived poor personal and professional life. How many allotted posts for surgeons remain vacant in the region? When a surgeon joins a remote location, how many days is he present on duty? (it is likely that many surgeons do not stay in their remote locations).

Response: Across the survey there was an average of 10% doctor (not surgeon) vacancies at the time of the survey. The main reason for doctors to leave is to acquire more training. Information on hours a doctor is present is not available. 

8. Time (and life) is lost in transit. Readiness can be assessed by how often a patient who should have been managed effectively is referred 'to a higher center'. What are the reasons for referral?

Response: Thank you for this important observation. The section on ‘Referrals’ (page 22-25, lines 331-382) and under the ‘Discussion’ (Pages 31-32, lines 507-523) discuss the issue of referrals and influencing factors.

9. Northeast India has large cities with large medical colleges and private hospitals. To generate data that represents the entire northeast, these should also be included in the study. 

Response: Thank you. Our sample was purposive since we focused on those organisations from which staff that joined the surgical training programme came. Hence, thereby, we aimed to provide insights into the challenges and opportunities for improving care. 

10. For the greatest impact on surgical care delivery, what are the recommendations for prudent investment in facility development, human resources and financial support?

Response: As indicated in the manuscript discussion, prudent investment in minimally invasive surgery including equipment and training as well as blood banking seem to be key.

Reviewer #5: Authors have satisfactorily reworked the manuscript. I just have one comment - in the map added to the revised version, J&K and Ladakh seemed to have cut-off. I am unsure if it is an issue with the PDF copy that was generated or something else. But it needs to corrected.

Response: Thank you for noting this. We have updated the map on page 6.

Reviewer #8: Thank you for the detailed responses to the comments and incorporating the suggested comments. It reads better now.

For the abstract results section, the quantitative results could be incorporated to support the statements mentioned. That would give a better perspective to the reader.

Kindly run the document through a spell check, there are places in the Introduction where 2015 is depicted as 2105,etc.

Response: We have updated the abstract on page 1-2. We have proof-read and made the necessary change on page 3. Thank you for picking this up, and bringing it to our attention.

---

## [Editor Report · Decision Letter 2]

15 Feb 2024

How ready is the health care system in North East India for surgical delivery? A mixed methods study on surgical capacity and need.

PONE-D-23-17522R2

Dear Dr. Virk,

We’re pleased to inform you that your manuscript has been judged scientifically suitable for publication and will be formally accepted for publication once it meets all outstanding technical requirements.

Kind regards,

Lovenish Bains, MS, FNB, FACS, FRCS (Glas), FICS, FIAGES

Academic Editor

PLOS ONE

Additional Editor Comments (optional):

Author responses are satisfactory.

Authors are advised to check the following for the country (India) specific IRB and Ethics committee for future.

https://www.naitik.gov.in/DHR/Homepage

https://www.naitik.gov.in/DHR/app_srv/soam/reports_pc_issued.jsp

This is the site maintained by ICMR (Govt of India) for IRB / Ethics Committee in the country and a unique number in the country is issued by them.